

# Tracing elevated abundance of $CH_2Cl_2$ in the subarctic upper troposphere to the Asian Summer Monsoon

Markus Jesswein[1], Valentin Lauther[3], Nicolas Emig[2], Peter Hoor[2], Timo Keber[1], Hans-Christoph Lachnitt[2], Linda Ort[2], Tanja Schuck[1], Johannes Strobel[3], Ronja Van Luijt[3], C. Michael Volk[3], Franziska Weyland[2], and Andreas Engel[1]

[1]University of Frankfurt, Institute for Atmospheric and Environmental Sciences, Frankfurt, Germany
[2]Johannes Gutenberg University of Mainz, Institute for Atmospheric Physics, Mainz, Germany
[3]Institute for Atmospheric and Environmental Research, University of Wuppertal, Wuppertal, Germany

**Correspondence:** Markus Jesswein (jesswein@iau.uni-frankfurt.de)

**Abstract.**

The Asian Summer Monsoon (ASM) is a seasonal weather pattern characterized by heavy rains and winds, mainly affecting South and Southeast Asia during the summer months. The deep convection within the ASM is an important transport process for pollutants from the planetary boundary layer up to the tropopause region. This study uses in situ observations of $CH_2Cl_2$ from the PHILEAS aircraft campaign in late summer 2023 to examine the transport pathways and timescales for polluted air from the ASM to the extratropical upper troposphere and lower stratosphere (UTLS). $CH_2Cl_2$ mixing ratios of up to 300 ppt ($\approx 500\%$ of the northern hemisphere background) were measured in the upper troposphere in the subarctic region. The largest observed pollution events were analyzed with the help of the Lagrangian particle dispersion model FLEXPART, both in terms of their origin and their potential entry into the lower stratosphere. The results show that the East Asia Summer Monsoon (EASM) is the key pathway for transporting uncontrolled Cl-VSLS into the tropopause region, which contributes to an increase in tropospheric background levels with the potential to enter the lower stratosphere. The transport analysis of elevated mixing ratios shown here suggests that transport to the upper troposphere in the subarctic region did not occur through transport into the Asian Summer Monsoon Anticyclone (ASMA) with subsequent eddy-shedding events, but rather by large convective transport contributions from the EASM. The projected entry into the lower stratosphere in the following days (12 days after the observation) amounts to a few percent. However, the analysis covered only a short time frame, suggesting that these elevated $CH_2Cl_2$ mixing ratios could still have the potential to enter the lower stratosphere at a later time.

## 1 Introduction

Short-lived chlorinated substances (Cl-VSLS) have shown increasing abundance in the atmosphere in recent years. These trace gases have local lifetimes less than half a year and are not controlled by the Montreal Protocol or its amendments and adjustments. The trace gases themselves and their product gases can reach the stratosphere and participate in catalytic cycles that destroy ozone with a general contribution depending on the spatial and temporal variability of their sources, transport pathways and chemical transformation (Laube and Tegtmeier, 2022). Model simulations conducted by Bednarz et al. (2023)



indicate that Cl-VSLS caused a decrease of approximately 2–3 DU (Dobson units) in the springtime at high latitudes during the second decade of the 21$^{st}$ century. Therefore, a continued increase in Cl-VSLS concentrations may postpone future recovery of the ozone layer (Hossaini et al., 2017).

Dichloromethane (CH$_2$Cl$_2$) is the most abundant Cl-VSLS in the atmosphere with mainly anthropogenic sources that make up around 90% of global emissions and a minor contribution from natural sources ( Montzka et al., 2010;  Laube and Tegtmeier, 2022). The atmospheric lifetime is estimated to be around 176 days (95–1070 days) (Burkholder et al., 2022). Between 2000 and 2020, global emissions increased by a factor of 2.5 to approximately 1.1 to 1.3 Tg yr$^{-1}$  (Laube and Tegtmeier, 2022).

Regional increases in emissions from the Asian region are the substantial part of the global increase (e.g. Claxton et al., 2020; An et al., 2021). In 2020, the global mean abundance of CH$_2$Cl$_2$ was around 40–45 ppt and thus nearly twice the amount compared to the early part of the century  (Laube and Tegtmeier, 2022).

The Asian Summer Monsoon (ASM) is a seasonal weather pattern characterized by heavy rains and winds, mainly affecting South and Southeast Asia during the summer months. Within the ASM region, deep convection rapidly transports air from the


planetary boundary layer (PBL) to the upper troposphere and lower stratosphere (UTLS) (e.g. Pan et al., 2016). Furthermore, the ASM forms a high pressure system in the UTLS, the Asian Summer Monsoon Anticyclone (ASMA), which acts as a transport barrier for uplifted pollutants (e.g. Park et al., 2007; Ploeger et al., 2015). However, the isolation of the ASMA is not perfect and there is, to some extent, horizontal exchange between ASMA and its surrounding by wave breaking or the so-called eddy shedding (e.g. Garny and Randel, 2016; Clemens et al., 2022).


The ASM itself consists of three subcomponents: the South Asian Summer Monsoon (SASM), the East Asian Summer Monsoon (EASM), and the Western North Pacific Summer Monsoon (WNPSM) (e.g. Ha et al., 2017). The latter constitutes an oceanic monsoon system. Thus, SASM and EASM are the two systems that influence continental areas with potentially polluted regions. The differential heating between the Indian and Pacific oceans and continental land masses governs SASM and EASM, with the Monsoon Trough serving as the main convergence zone in the SASM area and the East Asia subtropical


front in the EASM region (Pan et al., 2022, 2024). Li et al. (2022) examined the outputs from CMIP6 models to reveal the physical processes driving the unique changes in circulation in SASM and EASM due to global warming. Their findings consistently indicate a projected strengthening of the EASM circulation and a weakening of the SASM circulation in a future warmer climate.

Only a handful of airborne observations show the influence of ASM transport into the UTLS with respect to Cl-VSLS (e.g.


Oram et al., 2017; Adcock et al., 2021; Treadaway et al., 2022; Lauther et al., 2022; Pan et al., 2024). The most recent work by Pan et al. (2024) highlights EASM convection as an effective transport pathway for Cl-VSLS. They found extremely high values of CH$_2$Cl$_2$ (up to 600 ppt) in the region near the tropopause of East Asia during the ACCLIP campaign in August 2022. These substantial abundances arise due to the active deep EASM convection and the convergence zone directly over the primary Cl-VSLS emission sources (Pan et al., 2024).


This study uses in situ observations of CH$_2$Cl$_2$ from the PHILEAS aircraft campaign in 2023 to examine the transport pathways and timescales of CH$_2$Cl$_2$ from the ASM to the extratropical UTLS with the support of the Lagrangian particle dispersion model FLEXPART. Sect. 2 outlines the in situ measurements, the application of FLEXPART, and the analysis



of synoptic situations. In Sect. 3, we compare the observations of two in situ instruments and analyze selected events with extremely high CH$_2$Cl$_2$ in the upper troposphere during PHILEAS with respect to their origin and their possibility of further transport into the stratosphere. We briefly discuss our results in Sect. 4 and end with a summary and conclusion in Sect. 5.

## 2 Methods

### 2.1 The PHILEAS campaign 2023

The recent HALO campaign PHILEAS (Probing high latitude export of air from the Asian summer monsoon), undertaken in August and September 2023, had as one of its objectives the investigation of the primary transport pathways and time scales for polluted air from the Asian Summer Monsoon (ASM) into the extratropical upper troposphere and lower stratosphere (UTLS) (Riese et al., 2025, in preparation).

Throughout the PHILEAS campaign, a total of 20 flights were conducted, comprising 18 flights dedicated to scientific research and two flights aimed at calibrating turbulence measurements and assessing the electromagnetic compatibility between the instruments and the aircraft. The first phase took place in Oberpfaffenhofen, Germany, to investigate rather undiluted air from the ASMA and background lowermost stratospheric air. The second major phase was conducted from Anchorage, Alaska, tracing polluted air masses uplifted in the ASM and transported across the Pacific to the high-latitude UTLS. A final background flight was performed from Oberpfaffenhofen, with the two transfer flights between the two locations also serving as research flights. A more detailed description of all flights of the PHILEAS campaign is provided in the Table S 1 in the supporting information (SI). Furthermore, the flight tracks of the scientific flights are shown in Figures S 1 to S 3 in the SI.

### 2.2 in situ trace gas measurements

The HALO aircraft was equipped with a wide range of different in situ and remote sensing instruments (Riese et al., 2025, in preparation). For the analysis, only in situ observations of three instruments are used, which are described in the following sections. In addition to the scientific instruments, temporally installed for the campaign, the Basic HALO Measurements and Sensor System (BAHAMAS) is part of HALO. This permanently installed instrument provides meteorological and aircraft parameters along the flight tracks (Krautstrunk and Giez, 2012).

#### 2.2.1 CH$_2$Cl$_2$ observations

Dichloromethane (CH$_2$Cl$_2$) mixing ratios were measured using two distinct instruments on board the HALO research aircraft, the GhOST instrument from the University of Frankfurt and the HAGAR-V instrument from the University of Wuppertal. Details on the characteristics of these two instruments are provided in previous publications, including Jesswein et al. (2021) and Keber et al. (2020) for the GhOST instrument and Lauther et al. (2022) for a previous configuration of the HAGAR-V instrument. CH$_2$Cl$_2$ is measured with both instruments using gas chromatography (GC) and mass spectrometry (MS) with cryogenic sample preconcentration.





The GhOST includes a single GC-MS channel, sampling ambient air for 147 seconds with subsequent analysis, which leads to a time resolution of around 6 min per measurement cycle. Regular calibration measurements were performed during the research flights, and the measurement precision was derived for each flight from the standard deviation of these measurements. The average precision of $CH_2Cl_2$ throughout the PHILEAS campaign was 0.9%.

The HAGAR-V includes a two-channel GC with one MS. Using two GC systems can enhance the range of observable atmospheric trace gases or, by targeting the same species on both channels, doubles the frequency of observations (Lauther et al., 2022). The latter setup was selected during the PHILEAS campaign. With an ambient air sampling time of 30 seconds, a time resolution with the combined channels of 2 min is achieved. As for the GhOST, HAGAR-V is calibrated in-flight, but with two calibration gases. The average precision of $CH_2Cl_2$ throughout the PHILEAS campaign was 1.3%.

The calibration main gas bottles of both instruments, as well as the in-flight calibration gas cylinders, which were filled from the main gas bottles, were calibrated at the University of Frankfurt using a laboratory GC/MS (Schuck et al., 2018) or a Medusa (e.g. Miller et al., 2008; Arnold et al., 2012) system. The calibration relies on AGAGE-derived calibration according to the SIO-14 scale for $CH_2Cl_2$. All scientific flights were included in the analysis, except flight F02 due to malfunctions with the GhOST instrument for this flight.

### 2.2.2 N$_2$O observations

N$_2$O measurements were performed with the University of Mainz Airborne Quantum Cascade Laser Spectrometer (UMAQS). The instrument is based on direct absorption spectroscopy using a continuous-wave quantum cascade laser with a sweep rate of 2 kHz (Müller et al., 2015). The instrument consists of two units, each of which is equipped with a multipath cell (Herriott cell) with a path length of 76 m, operated at 40 Torr during PHILEAS. The instrument is calibrated in situ with two different working standards of compressed ambient air, which are compared to NOAA standards before and after the mission. Total uncertainty for N$_2$O is 0.08 ppb. The overall uncertainty can be obtained by adding the uncertainty for the working standards traceable to NOAA, which is 0.13 ppb for N$_2$O.

## 2.3 FLEXPART trajectories

Trajectory simulations were performed using version 11 of the Lagrangian particle dispersion model FLEXPART (Bakels et al., 2024). This is the newest version of the model, which was developed more than two decades ago (Stohl et al., 1998) with several improvements in between (e.g., Stohl et al., 2005; Pisso et al., 2019). Since then, this model has found application in numerous studies on atmospheric transport.

Central for our study is the effect of transport from the PBL and convection, including turbulence and subgrid winds. Therefore, we use FLEXPART, which utilizes motion vectors. These motion vectors combine the grid-scale wind velocity from linearly interpolated meteorological input data and the parameterized turbulent velocity. In addition, particles may be vertically displaced by convection (Bakels et al., 2024). FLEXPART accounts for subgrid-scale convection using the parameterization scheme of Emanuel and Živković Rothman (1999), which relies on the grid-scale temperature and humidity fields and calculates the convective displacement of the particles (Stohl et al., 2005). FLEXPART differentiates turbulence in the atmospheric



boundary layer (ABL) and turbulence in the free troposphere and stratosphere. Inside the ABL, turbulence is based on Hanna (1982) and Ryall and Maryon (1998), and a skewed turbulence option by Cassiani et al. (2015) (more information can be found in Bakels et al. (2024)). Above the ABL, turbulent velocities are computed following Legras et al. (2003). A constant vertical diffusivity is used in the stratosphere, whereas a horizontal diffusivity is used in the free troposphere. These diffusivities $(D_i)$

are converted into velocity scales using $\sigma_{vi} = \sqrt{2D_i/\Delta t}$ , where $i$ is the direction (Bakels et al., 2024).

The model is driven by the most recent ERA5 (Hersbach et al., 2020) hourly reanalysis data of the ECMWF (European Center for Medium-Range Forecasts), on a horizontal resolution of 0.5°x 0.5°. An important improvement of FLEXPART version 11 is the usage of the native vertical coordinates of ECMWF instead of interpolation to terrain-adapted coordinates with an improvement in trajectory accuracy and the better option for input and output of particle properties (Bakels et al.,

130    2024).

The particles are released in 5-minute intervals along the flight tracks with box sizes given by the latitude, longitude, and pressure ranges within the respective 5-minute interval. Loss processes due to deposition or chemical reactions are neglected, with transport being the focus only. Each release consists of 5000 computational particles (referred to as particles in the following), and trajectories of the particles are calculated for 12 days (forward and backward in time). The information related

to each particle is written as an hourly output, containing its position and meteorological information such as temperature, pressure, the height of the ABL, and potential vorticity, to name the most important ones for this study.

### 2.4    ERA5 reanalysis - climatologic and synoptic situation

We used ERA5 hourly data on single level and pressure levels (Copernicus Climate Change Service, 2018) to assess the climatological situation of the Asian Summer Monsoon (ASM) and the synoptic meteorological situation at the time of the

PHILEAS campaign. The focus is on the Northern Hemisphere from 0 to 180°E and pressure levels range from 850 hPa to up to 150 hPa with a horizontal resolution of 0.25°x 0.25°for the months of August and September 2023. Parameters such as convective available potential energy (CAPE) and total cloud cover were used. We derived several parameters from the ERA5 variables for the analysis of the meteorological situation.

The equivalent thickness is the vertical distance between two pressure levels. The thickness is related to the density and

temperature of the air, with decreasing thickness for colder air and increasing thickness for warmer air. Zones of high-thickness gradients help to identify fronts and boundaries between air masses.

Another parameter for frontal analysis is the Thermal Front Parameter (TFP) (Renard and Clarke, 1965; Hewson, 1998). The TFP describes the spatial change in the absolute value of the temperature gradient, but only the part of it that points in the direction of the temperature gradient. The mathematical definition is

$$TFP = \nabla \left| \nabla T \right| * \frac{\nabla T}{\left| \nabla T \right|} \tag{1}$$



with $T$ the temperature or the equivalent potential temperature and defining a threshold value to be exceeded to define a front, e.g. a TFP > 1 K $(100 \, \text{km})^{-2}$ (e.g. Kitabatake, 2008). The maximum of the TFP is located on the warm side of the zones with high thickness gradients. Thus, a combination of both parameters is well suited for the interpretation of front analyses.

The Q-Vector is an atmospheric dynamic parameter well suited for analyzing vertical motion in synoptic-scale weather
systems, first introduced by Hoskins et al. (1978). It is the rate of change of the horizontal temperature gradient, following the geostrophic flow. Included are changes in both magnitude and orientation. The Q-Vector is defined in Bluestein (1992) as follows

$$\boldsymbol{Q} = -\frac{R_d T}{\sigma} \left( \frac{\partial \boldsymbol{v_g}}{\partial x} \nabla T, \frac{\partial \boldsymbol{v_g}}{\partial y} \nabla T \right) \tag{2}$$

with $R_d$ the specific gas constant for dry air, $\sigma$ the static stability, and $\boldsymbol{v_g}$ the geostrophic wind. The Q-Vector form of the
omega equation (Eq. 3) can be helpful in finding areas of uplift and subsidence on a synoptic scale.

$$\left( \nabla^2 + \frac{f_0^2}{\sigma} \frac{\partial^2}{\partial p^2} \right) \omega = -2 \nabla \boldsymbol{Q} \tag{3}$$

$f_0$ in the equation is a constant Coriolis parameter. Thus, $-2\nabla \boldsymbol{Q}$ determines the forcing for vertical motions with $2\nabla \boldsymbol{Q} > 0$ ($2\nabla \boldsymbol{Q} < 0$) associated with sinking (rising) motion (e.g. Lackmann, 2011).

## 3 Results

### 3.1 Asian Summer Monsoon in 2023

In 2023, the Asian Summer Monsoon occurred under El Niño conditions after three consecutive years of La Niña. However, El Niño conditions did not fully manifest in the atmosphere and ocean until early September. The rainfall in the ASM region showed typical levels, but exhibited significant spatial and temporal variability. South Korea experienced higher than average precipitation. In China, the precipitation in June was below average, while it was above average in August and September
(World Meteorological Organization, 2024). Riese et al. (2025) (in preparation) employed the CLaMS model and multiple origin tracers, collectively referred to as a South Asia tracer, to place PHILEAS measurements within a climatological context, focusing on transport from the Asian Summer Monsoon. They show that in 2023, there was a slight northward displacement of the eastward outflow accompanied by a somewhat more intense than average flushing of the lowermost stratosphere.

### 3.2 Major observations of elevated CH$_2$Cl$_2$ in the upper troposphere

Dichloromethane mixing ratios were measured by two in situ instruments during the PHILEAS campaign. When comparing the measurements, the HAGAR-V instrument's higher time resolution captured multiple observations within a single measurement period of the GhOST instrument. Averaging HAGAR-V observations over the time periods corresponding to the enrichment



of the GhOST sample, as shown in Figure 1, reveals that the $CH_2Cl_2$ measurements from both instruments exhibit a slope of nearly 1 to 1, indicating excellent agreement, as confirmed by orthogonal distance regression (slope of 1.0074). We thus use observations of both instruments in the following.

Using the relationship of $CH_2Cl_2$ and $N_2O$ (Fig. 2 A), periods with considerably increased $CH_2Cl_2$ without similarly increasing $N_2O$ can be identified. $N_2O$, a long-lived greenhouse gas, has an atmospheric lifetime of approximately $116\pm9$ years (Prather et al., 2015). It is well mixed throughout the troposphere, and its primary removal mechanism is through photochemical destruction processes in the stratosphere. Furthermore, Figure 2 A shows the median polynomial fit and the 90th and 10th percentile polynomial fit (all polynomials of degree 3), within which 80% of the observations are located. For mixing ratios below approximately 330 ppb $N_2O$, a rather compact relationship is observed between $N_2O$ and $CH_2Cl_2$. Only during flight F07, several elevated mixing ratios were detected by both instruments. For $N_2O$ mixing ratios that are approximately above 330 ppb, a clear partitioning of very high and less pronounced lower values of $CH_2Cl_2$ can be recognized.

We used the median fit function of the $CH_2Cl_2$-$N_2O$ relationship to identify the major elevated observations during PHILEAS. For each $CH_2Cl_2$ observation, we derived a corresponding median $CH_2Cl_2$ mixing ratio ($CH_2Cl_{2med}$) using the respective $N_2O$ mixing ratio and the median fit function. To more effectively highlight observations that substantially diverge from the median, the deviation ($\Delta CH_2Cl_2$) is determined by subtracting $CH_2Cl_{2med}$ from the observed $CH_2Cl_2$ value for each observation. $\Delta CH_2Cl_2$ is displayed as letter-value plots for every flight in Figure 2 B. The plots originate from a central line representing the median. Moving outwards, each subsequent letter-layer encompasses half of the residual data, beginning with the 50th percentile that includes 50 % of the data. Following this, the subsequent two segments hold 25 % of the data, continuing this pattern until only outliers are evident. Letter-value plots provide more detailed information in the tails compared to box plots, but only where the letter values reliably represent the corresponding quantiles (Hofmann et al., 2011). Four flights stand out where $CH_2Cl_2$ mixing ratios considerably higher than 100 ppt above $CH_2Cl_{2med}$ were observed. These are the flights F08, F09, F16, and F17. Three more flights show $\Delta CH_2Cl_2$ between 50 and 100 ppt.

To determine the periods in which extremely high levels (elevated events) of $CH_2Cl_2$ were observed during the flights, we only consider observations that are above the mixing ratio derived from the fit of the 90th percentile curve and a factor of 2 above the median. In addition, an elevated event must have a duration of at least 10 minutes without any observation situated between elevated observations that is not itself considered elevated.

The $CH_2Cl_2$ time series of F08 and F17 are shown in Figure 3 A and Figure 4 A with elevated events shaded in red. This work focuses on these two flights. The $CH_2Cl_2$ time series with elevated events shaded from all four flights (F08, F09, F16, and F17) can be found in the supporting information (Figure S 4). F08 shows a similar high $CH_2Cl_2$ deviation to F16, but its letter-plot tailing towards larger deviations is slightly more pronounced (see Fig. 2 B). F17 shows the most pronounced deviation from the median and the overall largest mixing ratios of $CH_2Cl_2$ measured during the PHILEAS campaign. These two flights show three very clear events with $CH_2Cl_2$ mixing ratios of 200 to 300 ppt at an altitude of about 11 to 12.5 km and 330 to 350 K potential temperature ($\Theta$) across the northwestern Pacific to the subarctic region of Alaska (Fig. 3 B and 4 B). In the upper troposphere at comparable potential temperatures, PHILEAS observations within the 10th and 90th percentile ranged



from 50 ppt to 80 ppt. Figure 5 displays the flight tracks of these two flights with highlighted segments of the three elevated events.

### 3.3 Tracing back origin of high CH$_2$Cl$_2$

To further investigate the origins of the observed elevated CH$_2$Cl$_2$ mixing ratios, we utilize the Lagrangian particle dispersion model FLEXPART, running it in backward mode to trace the origins of air masses. Computational air particles, released on the flight path, were followed backward in time until they reached the PBL. Figures 3 C and 4 C display the median latitude and longitude (only 0–180°E to simplify presentation) of last PBL contacts of released particles within 5 min intervals along the flight tracks. The backward trajectories indicate a large variability in the median PBL contacts, and the CH$_2$Cl$_2$ mixing ratios

appear to be sensitive to the last PBL contacts observed. This suggests that air masses collected along the flight paths were influenced by various transport histories and were impacted by different source regions.

  The two major events of flight F08 exhibit differences in the locations of their most recent PBL contacts. The first event is associated with air masses originating in the longitudinal range of 100–115°E while the second event is linked to air masses from the 115–130°E region. Furthermore, the first event is more spatially constrained, occurring within the 30–40°N latitude

band, whereas the second event encompasses a broader latitude range of around 20–40°N. The event observed during flight F17 illustrates an internal structure of CH$_2$Cl$_2$ mixing ratios with a maximum occurring approximately between 02:10 and 02:20 UTC. The calculations of FLEXPART suggest that the last contact of these air masses with the PBL was located over a wide range of longitudes but was restricted to a rather narrow latitudinal band around 30°N.

  In both flights, the particles released within the 5-minute intervals in the shaded regions of Figure 3 and Figure 4 show

the highest relative proportion of their respective 5-minute intervals that have reached the PBL within the 12 day time of our FLEXPART calculation (see Figures S 5 and S 6 in the SI). More specifically, during the first event, the proportions of particle trajectories that have reached the PBL in the respective 5 min intervals are between 50% and 90%, while in the second event this figure ranges between 40% and 50%. In the F17 event, the relative share exceeds 40%, with occasional peaks between 60% and 80%, which could indicate mixed air masses of different transport pathways. On isentropic levels similar to those associated

with the three events, there exist time intervals during these flights where there is an increased percentage of trajectories that reach the PBL as well. However, these trajectories have lower percentages and different median latitudes and longitudes compared to the analyzed events. In the following, we examine the three major events in more depth.

### 3.3.1 Location of last PBL contacts and the maximum updraft along event trajectories

  For each event, we combine the particle trajectories from the 5-minute intervals within the event duration, as dictated by the

aforementioned conditions and represented by the red-shaded time ranges in Figures 3 and 4. The initial event occurred on August 26, 2023, from 19:25 to 20:05 UTC, featuring 40,000 particles. The following event on the same day spans from 20:25 to 21:15 UTC with 50,000 particles. The final event was observed on September 17, 2023, from 01:50 to 02:50 UTC, with 60,000 particles.



We not only looked at the last PBL contacts of each particle's path, but we also conducted an in-depth analysis of regions
exhibiting the highest diabatic ascent rates to pinpoint areas of notable updraft along the particles' trajectories during these
events. We followed the approach of Hanumanthu et al. (2020) and Lauther et al. (2022) to determine the maximum variation
in potential temperature for each particle trajectory over an 18-hour period and report the central coordinates for these particles.
The focus is on the spatial details of maximum rates along the backward trajectories of the particles rather than on their absolute
values. To achieve this, we implemented a rolling window to aggregate the hourly changes in the potential temperature. To better
illustrate the findings, the last PBL contacts and the maximum updraft within an 18-hour time frame ($\Delta\Theta_{18h}$) were binned into
$2° \times 2°$ latitude-longitude intervals.

During the first event on flight F08 on August 26, 2023, almost 80% of the 40,000 particles (combined 5-minute intervals)
reached the PBL during the 12 days. Most of the particle trajectories reached the PBL in the Huabei region of northern China,
as well as parts of northwest China, including Gansu, Ningxia, and Qinghai provinces. Furthermore, some pathways extended
to the far western part of northwest China, the tri-border area of Turkey, Iran, and Iraq, as well as the Pacific Ocean (Fig. 6 A).
Transport times to the observation location, particularly from northern China, range from approximately 5 to 6 days, with even
shorter transport times observed from trajectories originating in areas near the coast (Fig. 6 B). The most intense updrafts are
found in northern China near the coastal areas of Hebei, Beijing, and Shanxi (Fig. 6 C and D).

Following the first event on flight F08, the second event was observed soon after, albeit at different isentropic levels. The
pattern of the last PBL contacts and the maximum updraft differs somewhat from that of the first event, suggesting that different
mechanisms are responsible for the transport. Approximately 40% of the 50,000 particles reached the PBL during the 12 days
of calculation. The most recent PBL contacts are scattered across the Southeast and East Asia regions, as well as somewhat
extending into the Pacific Ocean. However, a greater number of trajectories make their last PBL contacts in the Huadong region
of East China and in Korea (Fig. 7 A) with transport times to the observation location ranging from approximately 5 to 7 days.
Two primary zones show the strongest updrafts along the trajectories: one dense area stretching from the Yangtze River Delta
across the Yellow Sea, Korea, and into Russia's Primorje region, and another less concentrated, extending from Kamchatka
southeastward over the Pacific Ocean (Fig. 7 C). Both regions exhibit comparable updraft intensities (Fig. 7 D).

The backward calculations for the event on flight F17 on September 17, 2023 displays another distinctive pattern of PBL
contacts and strongest updrafts, compared to the two events on August 26, 2023. Approximately 60% of the 60,000 particles
reached the PBL during the 12 days. Most of the particle trajectories reached the PBL in the more southerly regions of eastern
Asia, extending into southern and southeastern Asia. The most recent encounters are located in eastern Asia, particularly closer
to the eastern coastal regions, occurring around 5 to 7 days before the flight (Fig. 8 A and B). The strongest updraft along the
particle trajectories is predominantly confined to a narrow band that stretches from northern India along the southern border
of the Tibetan Plateau, passing through Sichuan, Chongqing, Hubei, Anhui, and Jiangsu, and extending to Korea, with the
intensity being more pronounced in the area around central to eastern China (Fig. 8 C and D).



### 3.3.2 Synoptic situations and vertical motions

We examined synoptic meteorological conditions that corresponded to the unique patterns of the back trajectories' last PBL contacts and maximum updrafts. ERA5 reanalysis was used to identify areas of frontal structures and vertical motion. The key tools included the equivalent thickness of 850 to 500 hPa, the thermal front parameter (TFP) at 700 hPa, and the Q-Vector
and Q-Vector divergence at 500 hPa. Furthermore, 500 hPa, 850 hPa, and 200 hPa geopotential heights were analyzed, together with parameters such as wind and convective available potential energy (CAPE).

ERA5 reanalysis data on pressure levels are provided on an hourly basis. For visualization purposes, only snapshots of the meteorological situation are given, e.g. times with large updrafts. The chosen times roughly coincide with the greatest updraft, based on time series of $\Delta\Theta$ (potential temperature variation along particle trajectories), visually represented using color codes
according to latitude, longitude, and particle density (refer to Fig. S 7 to S 9 in the SI). Meteorological fields for periods exhibiting the greatest updrafts are also available in the SI, together with airmass RGB satellite (EUMETSAT) images for the times of the snapshots. Airmass RGB utilizes two water vapor and one ozone absorption channel to differentiate between air masses and high-altitude multi-layered clouds, supporting the examination of dynamic atmospheric processes.

The backward particle trajectories for the first event on flight F08 reveal multiple distinct occurrences of strong updrafts,
the most substantial on August 22 and 23 (see Fig. S 7). Therefore, we examine two temporal snapshots as depicted in Figure 9. Figure 9 A presents a snapshot from August 22, 2023 10 UTC of the CAPE parameter. Located within the coordinates of approximately 110–118°E and 30–38°N is a region of enhanced CAPE, thus a region promoting strong and sustained upward air movement. The most substantial period of updraft is on August 23, 2023 19 UTC for which Figure 9 B shows the Q-Vector (arrows) and Q-Vector convergence (yellow to orange) and divergence (light to dark blue), together with the 500 hPa
geopotential (black lines). Positioned at the peak of the updraft along the backward trajectories (see Fig. 6 D) is a zone of vertical upward motion, highlighted by the convergence of the Q-Vector. This is located downstream of a slightly negatively tilted 500 hPa trough, which gradually shifted east over the next few hours (not shown here). Furthermore, this area is located in the entrance region of an anticyclonically curved jetstreak (the 200 hPa wind is depicted in a figure in the SI), which is related to the vertical upward movement in the troposphere. The closest RGB airmass satellite image at 18 UTC shows co-located
high-level thick clouds (Fig. S 10).

For the second event on flight F08, the greatest updraft, observed in most particles involved (Fig. 7 C), shows a distinct line from the Yangtze River Delta across the Yellow Sea, Korea, and into Russia's Primorje region. The peak period for the potential temperature gradient along the particles occurred from August 22 to early August 23, 2023 (see Fig. S 8). Since this appears to be a frontal structure, we utilize the thermal front parameter (TFP) for our investigation. Figure 10 A shows large
TFP values (yellow to red shadings), located on the warm side of the thickness crowding zone (gradient range of the black lines from shallow to high thickness). This structure remained almost stationary from 22 August 2023 until the early hours of August 23, 2023. The Q-Vector convergence implies upward vertical motion for the region of the Yellow Sea, Korea, and Russia's Primorje region (Fig. 10 B). Moreover, the 500 hPa geopotential chart reveals a detached upper-level low at around 120°E and 50°N. Consequently, this collectively indicates the potential presence of a frontal structure in this area. The RGB



airmass satellite image supports this by showing a high-level thick band of clouds along the frontal zone with dark blue to
brown colors (cold and dry air) on the rear side (Fig. S 11).

The event of flight F17 shows again a distinct line of maximum updraft from Korea, across China, and the southern border of
the Tibetan Plateau (see Fig. 8 C and D). It is the only one of the three events that indicates an updraft in the SASM Monsoon
Trough region and in the EASM. The Monsoon Trough region is accompanied by high CAPE values (see meteorological
charts in SI). Although strong updraft events occur at several times during the period of the backward calculation, a substantial
proportion of particles show their maximum upwind position on a very narrow line within 3 to 6 days prior to their release (see
Fig 8 C and S 9). Aligned with the strongest updraft area in east China and Korea are large values of the TFP (Fig. 11 A), which
begin to accumulate on September 12 and dissipate in the early hours of September 13, 2023. Fig 11 B illustrates consistent
areas of upward vertical motion (Q-Vector convergence) near the positively tilted trough in that time. Much like the second
event in flight F08, this frontal structure remains largely stationary throughout its duration. RGB airmass satellite image from
September 12, 2024 at 21 UTC reveals a thick cloud band, co-located to large values of TFP.

## 3.4 Projected contribution to the stratosphere

All three events were observed in the upper troposphere around 330 to 350 K of potential temperature, in proximity to the
tropopause and with tropospheric $N_2O$ values (e.g. Fig. 3 A, 4 A, and S. 13). To find out whether these air masses with high
$CH_2Cl_2$ can reach further into the stratosphere in the upcoming days, forward particle trajectories were generated. The proce-
dure mirrored that of the backward trajectories, employing 5-minute intervals along the flight paths and combining intervals
during the event durations. We classify the particle trajectories into three regions based on potential vorticity (potential vorticity
unit; PVU). A specific PVU value for defining the dynamical tropopause is not universally established, but 2 PVU is a fre-
quently used threshold (e.g. Holton et al., 1995). Kunz et al. (2009) described the region where mixing occurs in the tropopause
region as the area that encompasses the dynamic tropopause and thermal tropopause, termed the "Tropospheric Freshly Mixed
(TFM) branch". Furthermore, the lower limit of the TFM branch is established by the dynamical tropopause at 2 PVU, while
the mean potential vorticity at the thermal tropopause is estimated to be around 4 PVU. Taking this into account, we utilize
the following classification: The first region is assigned to potential vorticity values below 2 PVU, indicating that it is in the
troposphere. The second region is assigned to values ranging from 2 to 4 PVU, which identifies it as the tropopause region.
The third region, with values exceeding 4 PVU, is classified as stratospheric. Table 1 shows the percentage share of the ranges
2-4 PVU and greater than 4 PVU (the remaining percentage is below 2 PVU), averaged over 48-hour intervals.

All three events demonstrate comparable slight increases in the stratospheric portion (> 4 PVU) and exhibit more variations
in the share within the tropopause region (2–4 PVU). In addition, the first event on flight F08 is the only one that shows a minor
presence of particles in the tropopause region and stratosphere within the first 48 h after release.

The first event on flight F08 indicates a minor presence of the particles in the tropopause region with a decrease within the
following days. For the second event on flight F08, the largest share in the tropopause region was within 7 to 8 days after
release with up to 8.9% with a decreasing contribution afterwards, probably due to the return of some particles to the free





troposphere. The largest share to the tropopause region can be seen for the event on fight F17 with up to 27.6% on days 5 and 6 with a decreased share in the following days.

The proportion in the stratosphere increases within the 12 days of forward calculation to around 2.4% and 3.8% for the three events. For the event on flight F08, the stratospheric proportion exists from the beginning and peaks at 3.8% after 5 to 6 days. In contrast, for the second event on flight F08 and the event on flight F17, the stratospheric share starts 5 to 6 days after release, reaching its peaks at 2. 4% and 3. 2%, respectively, after 11 to 12 days. The steady increase of particles in the stratosphere within the 12 days could possibly continue in the following days.

**4   Discussion**

$CH_2Cl_2$ mixing ratios of up to 300 ppt were observed in the upper troposphere (11–12.5 km) above Alaska and the Gulf of Alaska region. For comparison, observations during take-off and landing at Anchorage show $CH_2Cl_2$ mixing ratios of about 60 ppt near the ground (see Fig. 3 and 4). We were able to assign the origin of these large mixing ratios to the region of the ASM. We were not the first to observe elevated mixing ratios of $CH_2Cl_2$ in the upper troposphere, although the focus of

previous studies was predominantly on measurements within the ASM or transport in a tropical direction. For instance, Oram et al. (2017) detected $CH_2Cl_2$ mixing ratios as high as 121 ppt in the upper troposphere (10–12 km) over the Bay of Bengal, originating from East Asia and with potential for transport into the tropical regions of the western Pacific, eventually rising to the tropical upper troposphere. Treadaway et al. (2022) investigated the transport of Asian emissions to the tropical tropopause layer of the West Pacific with a plume of around 90 ppt $CH_2Cl_2$ (at 14–16 km) with air that originated predominantly from

India. Adcock et al. (2021) found tropopause mixing ratios of $CH_2Cl_2$ in the range of 65–136 ppt in the tropopause region and indicated possible source regions in South Asia. All these studies show enhanced values of $CH_2Cl_2$ in the upper troposphere and tropopause region, but do not reach the values we observed in the upper troposphere in the subarctic area of Alaska. Furthermore, the focus these studies is on the tropical uplift of elevated $CH_2Cl_2$ and potential sources from India and South to East Asia. This study mainly examines elevated $CH_2Cl_2$ mixing ratios originating in the EASM region, which were transported

to higher latitudes over longer distances. Caution is advised when comparing the absolute values of $CH_2Cl_2$ with earlier studies, as there has been a notable increase in $CH_2Cl_2$ emissions over the past decade, rising by a factor of 2.5 (Laube and Tegtmeier, 2022). However, it is clear that the increase in emissions in the region of the EASM is developing much more strongly with considerable importance for transport into the upper troposphere and lower stratosphere.

    A more recent study by Pan et al. (2024) not only provides additional evidence for SASM injection of short-lived ozone

depleting substances, but also highlights the key role of EASM in injection into the stratosphere. During the ACCLIP campaign, an unusual northward shift of the convergence zone was observed due to an atypical configuration of the ASM for that year. Typically, the convergence zone is positioned slightly more southward. However, both in 2022 and climatologically, the convergence zone intersects with significant Cl-VSLS sources along the coastal regions of East Asia (Pan et al., 2024).

    Our findings support this statement, as our backward trajectories for three substantial $CH_2Cl_2$ plume observations in the

upper troposphere revealed the last PBL contacts in known source areas of $CH_2Cl_2$ and peak updrafts suggest predominantly



typical line structures, which can be associated with frontal structures and convergence zones (also visible in RGB airmass satellite images in the SI). This highlights once more the importance of viewing the EASM as a key pathway for transporting Cl-VSLS into the upper troposphere, thereby contributing to an increase in tropospheric background levels with the potential to enter the lower stratosphere. Moreover, the transport times and areas of increased $CH_2Cl_2$ mixing ratios shown here suggest that

transport to the upper troposphere in the subarctic region is driven by large convective transport contributions from the EASM. Previous studies have focused more on the entry of polluted air into the UTLS via the ASMA with subsequent eddy shedding events. The ASMA covers a range of potential temperatures from approximately 360–450 K (see, for example, Vogel et al., 2019). Small-scale eddies are shed from the main anticyclone (i.e. the so-called eddy shedding events) with quasi-horizontal isentropic transport out of the anticyclone, either directly into the lower stratosphere or into the tropical troposphere with

subsequent slower transport into the stratosphere (e.g. Clemens et al., 2022). Elevated $CH_2Cl_2$ mixing ratios were identified in a potential temperature range of 330–350 K in this study, thus below the ASMA.

Typical frontal structures in the EASM region include the Meiyu, Baiu, and Changma fronts, where the name changes with the location of the frontal structure. As the monsoon progresses, the structures shift northward. They first appear over Taiwan, southern China, and the Okinawa region from early May to mid-June. They then move to the Yangtze River valley and the

main islands of Japan from mid-June to mid-July, and finally reach the Korean Peninsula and northeast China during mid-July to mid-August (e.g. Jun-Mei et al., 2013). In this study, frontal structures were observed at the end of August. Shin et al. (2022) investigated the synoptic characteristics of the quasi-stationary front of August 26-27 2018 over the Korean peninsula, similar to the frontal structure within this study. The quasi-stationary front observed in their study exhibited features similar to the Changma front. The environmental conditions of the August event examined were atypical for heavy rainfall with a quasi-

stationary front and are closely related to the expansion of the subtropical high of the west Pacific (WPSH) (Shin et al., 2022). This study also observed an expansion of the WPSH, occurring not just in late August but also in mid-September accompanied by frontal structures.

We conducted a preliminary analysis of the potential for elevated $CH_2Cl_2$ occurrences to penetrate deeper into the tropopause region and lower stratosphere regions. The ongoing challenge is to mitigate stratospheric ozone depletion caused by the in-

creasing trend of uncontrolled Cl-VSLS (Chipperfield and Bekki, 2024). The forward trajectory analysis indicates varied contributions for the three events. For the tropopause region, the contribution fluctuated significantly between events, reaching approximately 25% during the largest event. The first event showed the largest overall contribution to the lower stratosphere, with values up to 3. 8%, while the other two events exhibited successively increasing contribution to the lowermost stratosphere up to about 2-3% over the 12-day period. Although the influence on the stratosphere during these periods is relatively minor,

further transport into the lowermost stratosphere may take place over a longer duration. For instance, this can occur in the ridges of baroclinic waves on the anticyclonic side of the jet stream, situated above the outflow of warm conveyor belts (e.g. Kunkel et al., 2019).





## 5 Summary and Conclusion

We report on measurements of $CH_2Cl_2$ from two in situ instruments and $N_2O$ from a third in situ instrument during the HALO
aircraft campaign PHILEAS in late summer 2023. One of the primary scientific interests centered on how polluted air from
the Asian Summer Monsoon region reached the extratropical upper troposphere and lower stratosphere (UTLS), with research
flights mainly originating from Anchorage, Alaska. In addition, the FLEXPART Lagrangian dispersion model was used to
investigate the origin of selected pollution events, the corresponding transport times from the planetary boundary layer (PBL),
and the potential for further input into the lower stratosphere, using calculations extending 12 days both backward and forward.

The measurements of $CH_2Cl_2$ recorded by the two in situ instruments align very well. Major pollution events during the
PHILEAS campaign were identified by examining the relationship between $CH_2Cl_2$ and $N_2O$. Two flights (F08 and F17)
showed three very clear events with $CH_2Cl_2$ mixing ratios of 200 to 300 ppt at altitudes of about 11 to 12.5 km and 330 to
350 K potential temperature over the northwestern Pacific up to the subarctic region of Alaska. These mixing ratios deviate
substantially from the comparable median mixing ratios of the campaign.

The FLEXPART model in backward mode was used to trace the origins and transport times of air masses responsible for
elevated mixing ratios observed during the aforementioned pollution events. For each event, the paths of the computational air
particles were analyzed to identify the last planetary boundary layer (PBL) contacts and regions of maximum diabatic ascent
using potential temperature variations along particle trajectories. Transport times between PBL and the flight paths for the three
events were around 5-7 days. For the first event in flight F08, the particle trajectories reached the PBL primarily over northern
China (approximately 100-115° E and 30-40° N), with updrafts focused closer to coastal areas. For the second event in flight
F08, the particle trajectories, which reached the PBL, dispersed across Southeast and East Asia (approximately 115-130° E
and 20-40° N). The updrafts were notably strong along the Yangtze River Delta, Korea, and extending into Russia. For the
event on flight F17, the particle trajectories reached the PBL in broader region of southern to southeastern Asia (approximately
80-120° E and 20-30° N). Updrafts were concentrated from northern India to eastern China and Korea.

ERA5 reanalysis was used to identify meteorological conditions, such as frontal structures and vertical motions that aligned
with particle trajectory patterns and updraft intensities. Key findings indicated distinct patterns in each event, influenced by
regional meteorological conditions such as convective areas and frontal structures, revealed through parameters such as CAPE,
Q-Vector divergence, and the thermal front parameter (TFP). RGB airmass satellite images were used to confirm the presence
of frontal structure with associated cloud formations.

This analysis of the elevated $CH_2Cl_2$ mixing ratios indicates that the transport into the upper troposphere in the subarctic
region resulted from convective upwelling by the EASM and subsequent displacement over the northern Pacific Ocean before
the air masses could ascent further into the ASMA and were subsequently transported into the lower stratosphere.

By forward FLEXPART calculation of the three events, the potential of these observed elevated $CH_2Cl_2$ mixing ratios to
reach the lower stratosphere in the following days (12-day time period) was investigated. The estimated entry into the lower
stratosphere over the coming days is only a few percent (1.3% to 3.8%). However, the study was based on a limited time frame,



indicating that elevated $CH_2Cl_2$ mixing ratios may still have the opportunity to reach the lower stratosphere in a longer time range.

*Data availability.* The observational data of the HALO flights during the PHILEAS campaign are available via the HALO database (https://halo-db.pa.op.dlr.de) or upon request from the main author. The calculation of the FLEXPART model is available upon request.

*Author contributions.* MJ, TK, TS, and AE operated and provided data of the GhOST instrument; VL, RVL, JS, and CMV operated and provided data of the HAGAR-V instrument; NE, FW, HCL, and PH operated and provided data of the UMAQS instrument; MJ did the FLEXPART simulations. MJ performed the data analysis and wrote the paper. All authors have contributed via discussions and comments.

*Competing interests.* At least one of the (co-)authors is a member of the editorial board of Atmospheric Chemistry and Physics

*Acknowledgements.* This work was done at the University of Frankfurt. The authors thank the DLR staff for the operation of the HALO and
the support during the campaign, and also the coordinators and colleagues for productive cooperation during the campaign.





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



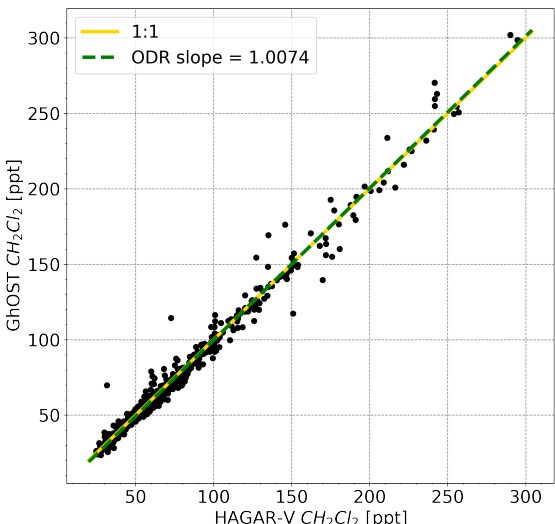

**Figure 1.** Correlation of $CH_2Cl_2$ measured with the HAGAR-V instrument (x-axis) and the GhOST instrument (y-axis). HAGAR-V measurements were averaged when more than one observation is within the sample time of the GhOST instrument. The 1-to-1 line is shown in red (solid) and the slope of the orthogonal distance regression is shown in green (dashed)

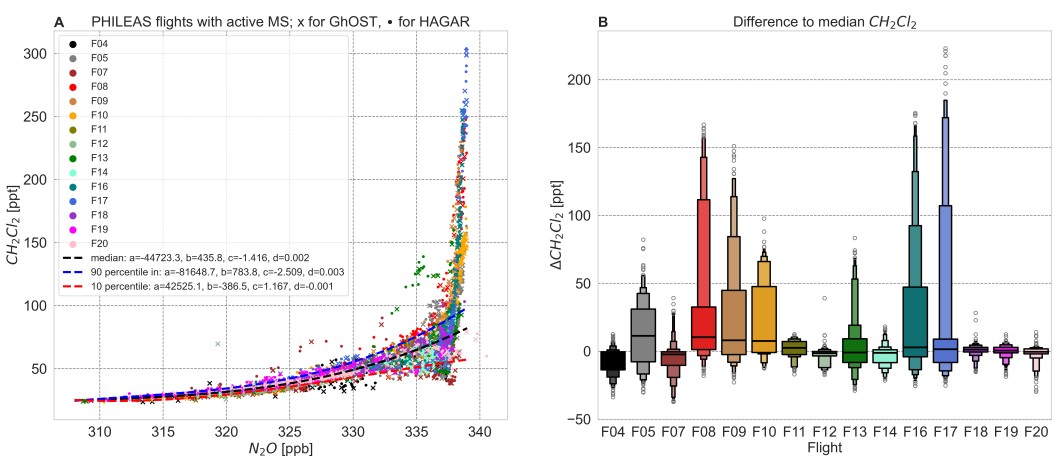

**Figure 2. A** $CH_2Cl_2$-$N_2O$ relationship, color coded by flight. Crosses for HAGAR-V and points for GhOST. $N_2O$ is averaged to the respective sample enrichment time of the gas chromatographs. Black dashed curve indicates median curve fit, blue and red dashed line the $90^{th}$ and $10^{th}$ percentile curve fits. **B** Letter-value plots of $\Delta CH_2Cl_2$ for every PHILEAS flight to median $CH_2Cl_2$ value derived from median curve fit. Horizontal lines show median and points extreme outliers.



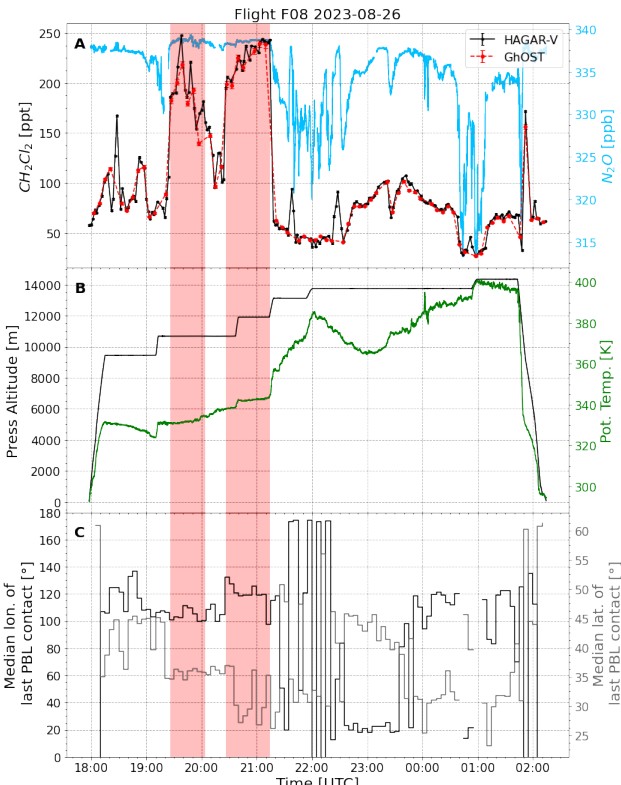

**Figure 3. A** Time series of flight F08 on August 26, 2023. Mixing ratios of CH$_2$Cl$_2$ of the GhOST in red and HAGAR-V in black (left y-axis) and N$_2$O in blue (right y-axis). **B** Pressure altitude in black and potential temperature in green along the flight path. **C** FLEXPART median latitude in grey and longitude in black of particles within the 5-min intervals. Major elevated time ranges are highlighted in red.



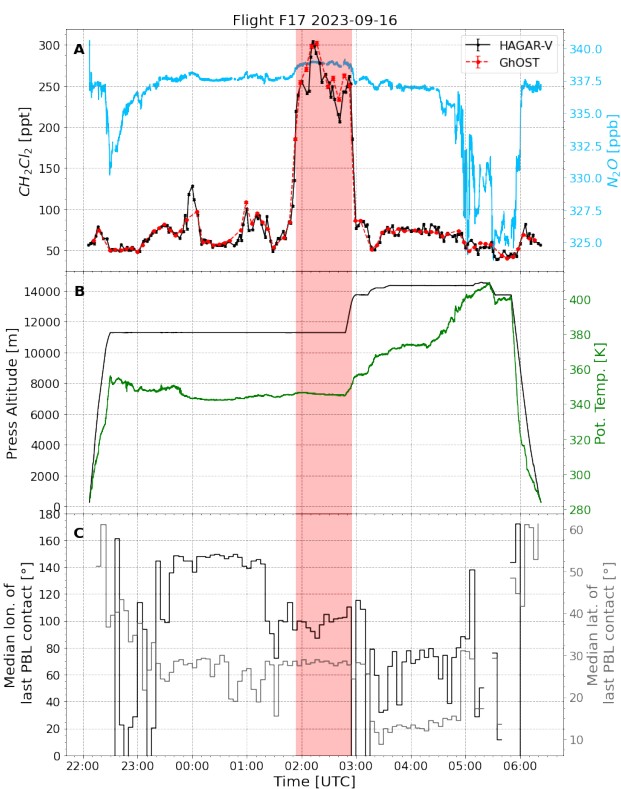

**Figure 4.** like Fig. 3 but for flight F17 on September 16, 2023.



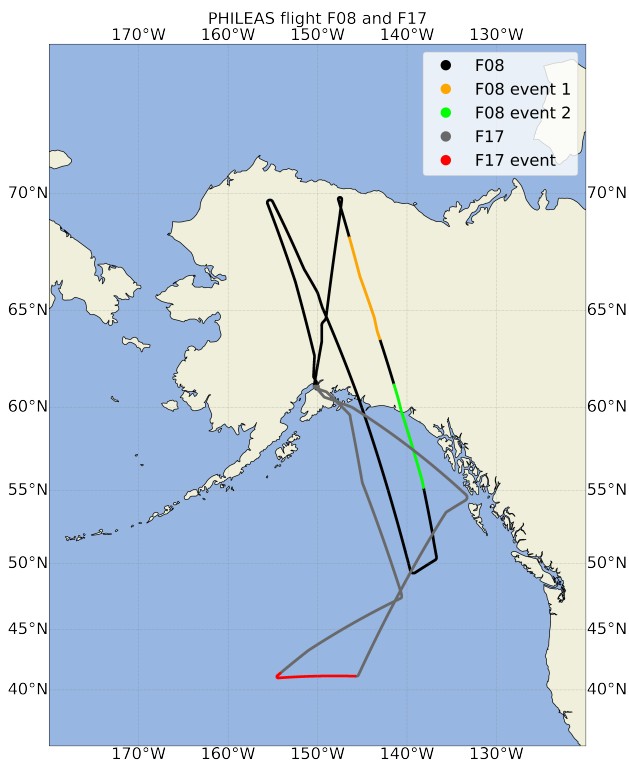

**Figure 5.** Flight tracks of flight F08 on August 26 (black) and flight F17 on September 16, 2023. Flight segments with high values of $CH_2Cl_2$ were highlighted in orange, green, and red (F08 event 1, F08 event 2, and F17 event, respectively).





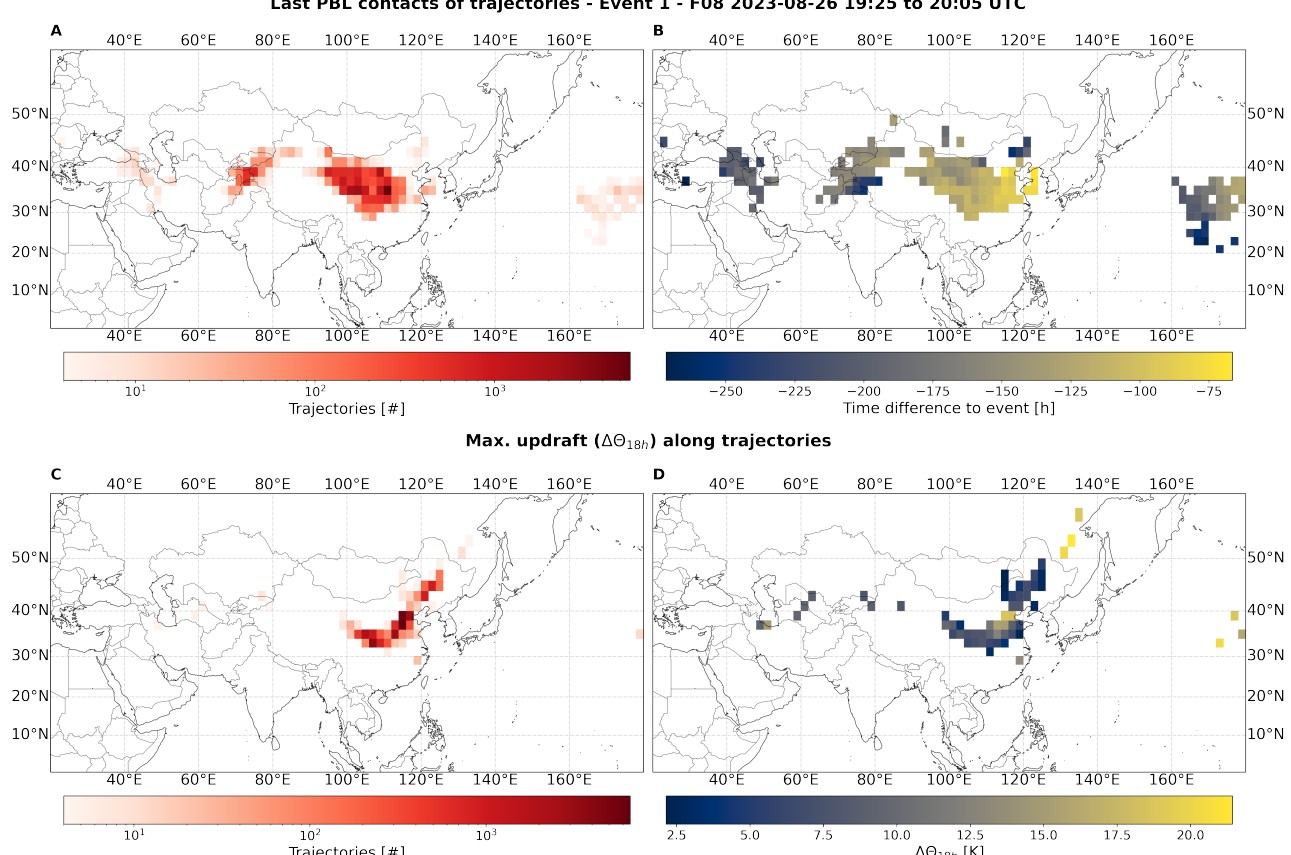

**Figure 6.** Averaged over $2.0° \times 2.0°$ latitude-longitude intervals for the first event of flight F08, the last PBL contacts and maximum updrafts along the trajectories are shown. Panels **A** and **C** illustrate these intervals, color-coded by the logarithmic scale of trajectory densities for last PBL contacts and maximum updrafts. Panel **B** displays time differences to release time, using color-coded intervals in hours. Panel **D** displays the location of max. updraft within 18 h along the particle trajectories, color-coded by potential temperature difference within the 18 h



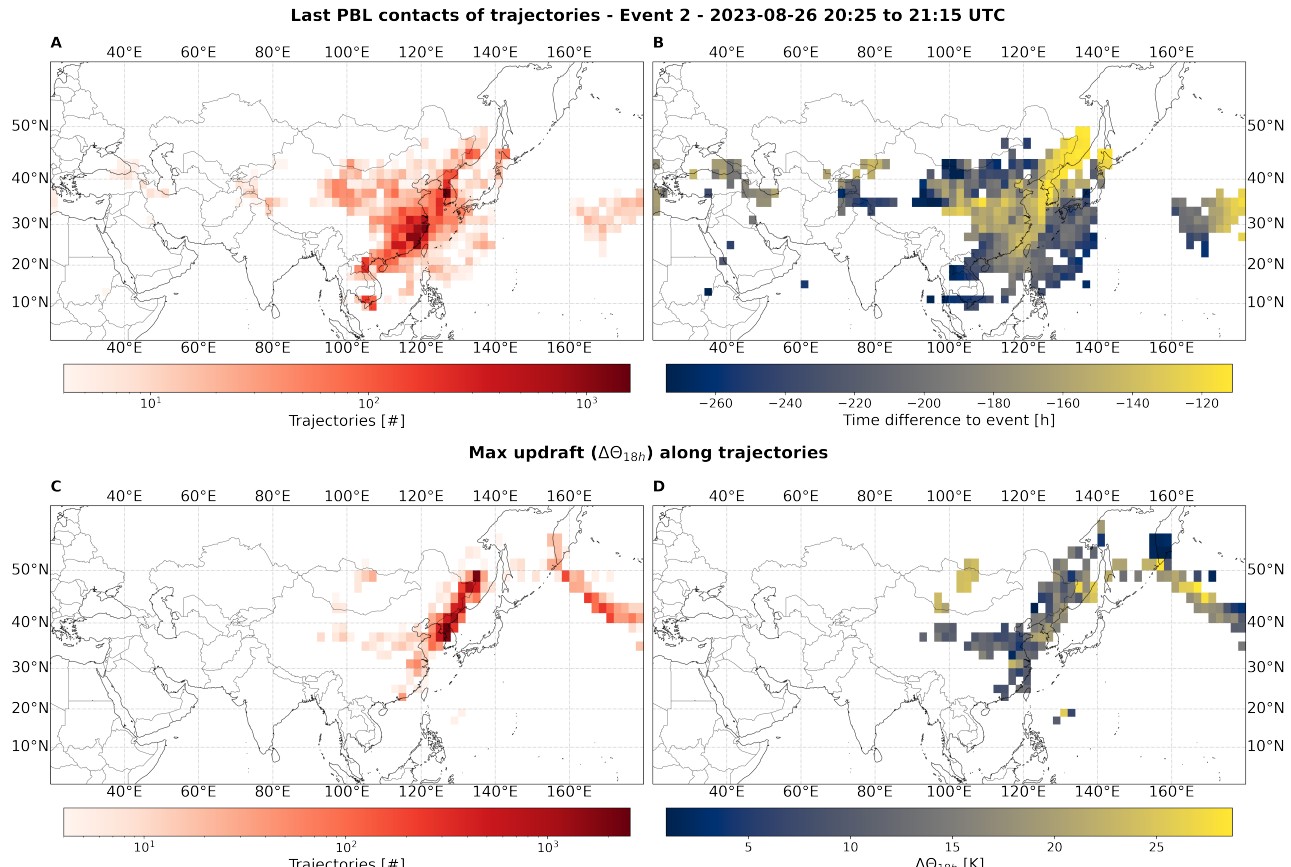

**Figure 7.** like Fig. 6, but for the second event on flight F08.



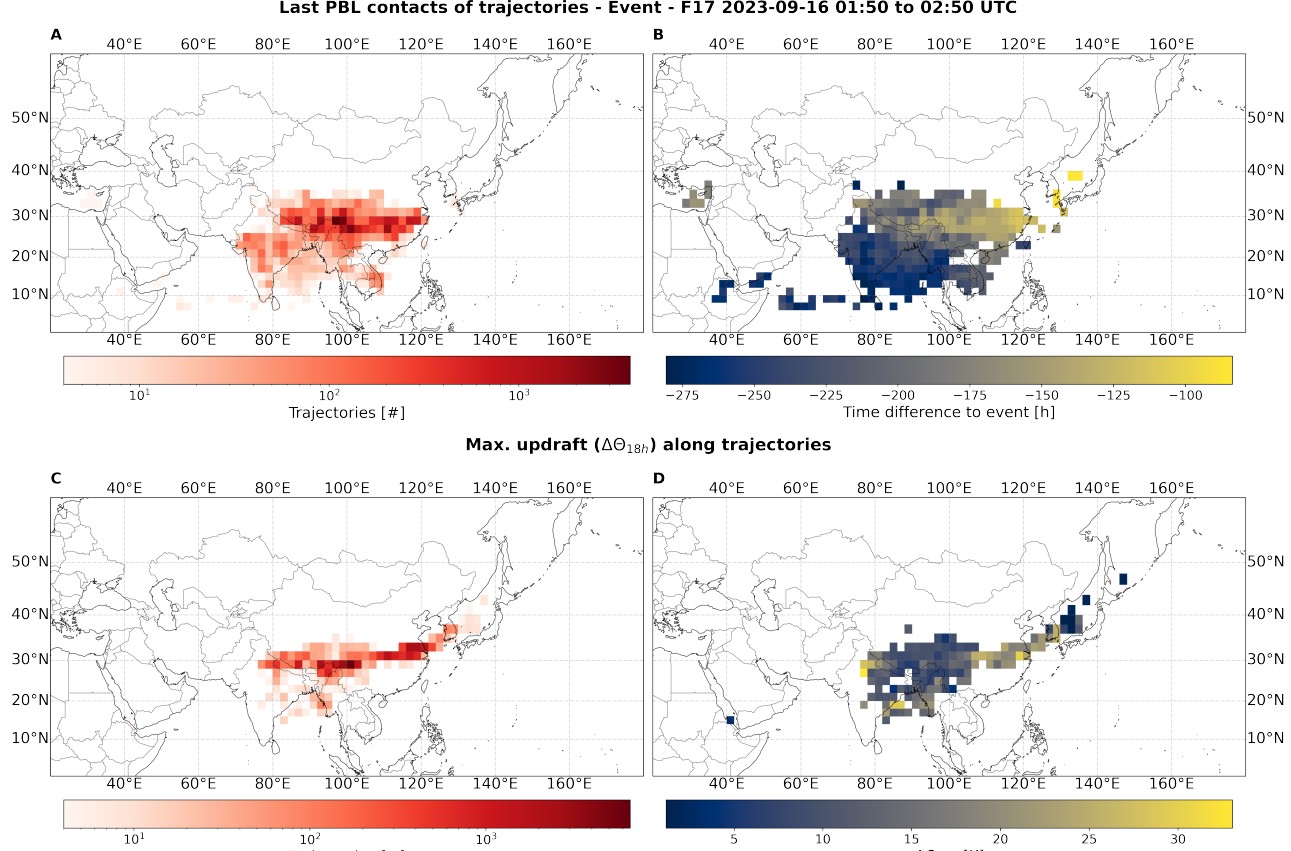

**Figure 8.** like Fig. 6, but for the event on flight F17.





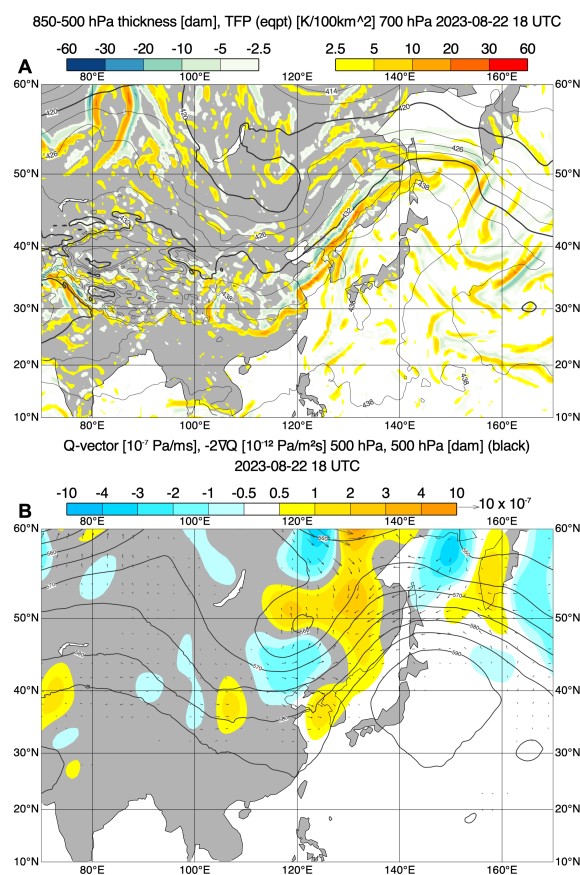

**Figure 10.** Analyse maps from 2023-08-22 18 UTC. **A** shows the thermal front parameter (TFP) and 850–500 hPa thickness. **B** shows Q-Vector, Q-Vector vergences, and 500 hPa geopotential.



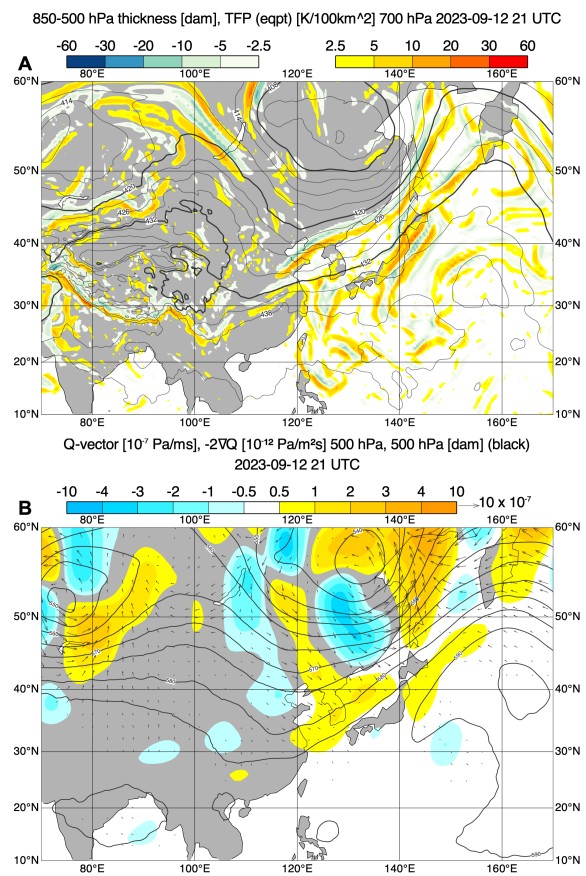

**Figure 11.** like Fig. 10, but for 2023-09-12 21 UTC.





**Table 1.** Percentage (%) of particle trajectories that are in a potential vorticity range of 2-4 PVU (tropopause region) or greater than 4 PVU (stratospheric). Given in 48 hour averages along the forward trajectories.

|  |  | $\Delta$48h | $\Delta$96h | $\Delta$144h | $\Delta$192h | $\Delta$240h | $\Delta$288h |
|---|---|---|---|---|---|---|---|
| F08 event 1 | 2-4 PVU | 3.8 | 3.8 | 2.6 | 1.9 | 1.9 | 1.7 |
|  | >4 PVU | 1.5 | 3.5 | 3.8 | 3.4 | 3.8 | 3.7 |
| F08 event 2 | 2-4 PVU | 0.0 | 0.1 | 6.3 | 8.9 | 3.3 | 2.6 |
|  | >4 PVU | 0.0 | 0.0 | 0.3 | 1.0 | 1.5 | 2.4 |
| F17 event | 2-4 PVU | 0.0 | 5.2 | 27.6 | 12.0 | 11.8 | 12.3 |
|  | >4 PVU | 0.0 | 0.0 | 0.6 | 1.3 | 1.5 | 3.2 |