# Peer review of "Tracing elevated abundance of $CH_2Cl_2$ in the subarctic upper troposphere to the Asian Summer Monsoon"

_EGUsphere, 2024_

## Author Comment (AC1)

**Jesswein et al., Tracing elevated abundances of CH$_2$Cl$_2$ in the subarctic upper troposphere to the Asian Summer Monsoon**

**Responses to comments by Anonymous Referee #1**

We thank the anonymous reviewer for the thorough review of the manuscript and the suggestions for improvement. All comments are addressed below, with the reviewer's comments in black and responses in blue with improvements listed in italics.

Comments:

Figure 1: The 1:1 line is yellow not red in the Figure.

Done.

Overall, figures with light colored axes are hard to read. An example is the light blue for Figure S5, perhaps a slightly darker shade of the color at least for the axes and label.

We have replaced the light colors in these figures with slightly darker ones so that the contrast is stronger.

In the supplement could you add figures showing the CH2Cl2 as a function of altitude or POT? It would be great to see the data visualized in this way as well to help readers comparing to recent literature work.

We included such a plot as Figure S4 in the supporting information and the reference to it can be found in Section 3.2.

Body of Text:

Line 67: It would be nice either here or the supplement to list the altitude or POT range covered by the flights.

The potential temperaure and altitude range of the research flights were included in this section.

Line 83: Would be nice to have the instrument names spelled out the first time it is used for the reader to know.

We included the instrument names in parentheses behind the abbreviation.

Line 321: 2023 and not 2024

Done.

Section 3.4: This analysis is really interesting. It would be great to see a rough estimate of the CH2Cl2 chlorine injection for PVU>4 to bring it back to the ozone assessment information.

The section on the stratospheric contribution is merely an outlook and is not the focus of this manuscript. This would require a more in-depth investigation beyond this case study. Therefore, elaboration of the stratospheric contribution with estimated $CH_2Cl_2$ chlorine injection would go beyond the scope of this manuscript and is not discussed in more detail.

---

## Author Comment (AC2)

**Jesswein et al., Tracing elevated abundances of $CH_2Cl_2$ in the subarctic upper troposphere to the Asian Summer Monsoon**

**Responses to comments by Anonymous Referee #2**

We thank the anonymous reviewer for the thorough review of the manuscript and the suggestions for improvement. All comments are addressed below, with the reviewer's comments in black and responses in blue with improvements listed in italics.

**General Remarks:**

- I think a little more work can be done to justify and explain the analysis road map throughout the paper. There is one remark in the introduction about how specific events will be analyzed, but it isn't clear until lines 204-205 that the majority of this paper will only discuss three flight segments from the entire campaign. Moreover, the analysis of Figure 2 reveals four flights of interest, but there doesn't appear to be any justification for the subsequent elimination of two of them (F09 and F16) from further consideration. I recommend providing a little more early clarity about the scope of this work, as well as further justification that the chosen three flight segments are enough to serve the overall goals of the study.

  We thank the reviewer for this comment regarding a more clear road map of the analysis of this manuscript. The first minor additon can be found in the abstract where we make clear that we look at the largest three events in this manuscript (inlcuding the word three). To make it more clear in the introduction, we modified the short description of section 3:

  *"In Sect. 3, we compare the observations of two in situ instruments. Furthermore, an analysis of the three events that show the highest $CH_2Cl_2$ mixing ratios in the upper troposphere during the PHILEAS campaign is carried out, focusing on their origins and potential for further transport into the stratosphere."*

  Indeed, Figure 2 reveals four flights with substantially larger values of $CH_2Cl_2$. Including all four flights (seven events in total) would make the manuscript and especially the discussion of the individual events difficult to follow. We have therefore reduced them to the three most important events. A justification for the choice of F08 and F17 can be found in lines 206-210 of the original manuscript, where the letter-plot tailing justifies the choice of F08 over F16, with F17 undoubtedly being the longest event during the campaign. The omission of flight F09 was not included in the text. To justify this, we have expanded this text passage.

*"The events in F09 show less coherent structures compared to those of the other three flights. F08 shows a similar high $CH_2Cl_2$ deviation to F16, but its letter-plot tailing towards larger deviations is slightly more pronounced (see Fig. 2 B). F17 shows the most pronounced deviation from the median and the overall largest mixing ratios of $CH_2Cl_2$ measured during the PHILEAS campaign. These two flights show three very clear events with $CH_2Cl_2$ mixing ratios of 200 to 300 ppt [...]"*

- I am not totally satisfied with the justifications provided for the relatively minor contribution of dichloromethane enhancements to the lowermost stratosphere. The authors clarify that the study spans a short time period, and that there could be a higher contribution after 12 days. However it is clear from Table 1 that most of the particles that reach 2-4 PVU do not eventually cross the 4 PVU threshold, and I would only expect the likelihood of that to decrease with additional time. The fact that the sampling was primarily below the ASMA (at 330-350K, as stated in lines 385-386) could be a simple way to justify this – it's not that the ASM doesn't have impacts on stratospheric composition, it's just that they are not as pronounced for air masses that reach the subarctic upper troposphere. With all this in mind, I suggest changing the tone of the discussion and conclusion (including the final two sentences of the abstract) to emphasize that the enhanced levels of dichloromethane observed during these specific segments had only a minor impact on the composition of the lower stratosphere based on the applied modeling approach.

  We have taken the last sentence of the abstract from the manuscript and expanded the previous sentence so that the tone is clearer that the contribution to the lower stratosphere is minor.

  *"The projected entry into the lower stratosphere in the following days amounts to a few percent, indicating that the direct influence of these particular events on the lower stratosphere is probably minor."*

  In the discussion, we further worte, that the contributions to the stratosphere from the three events are minor:

  *"All three events show a minor contribution to the lower stratosphere up to about 3.8% within the 12-day period following the events. We are considering a specific time frame and geographic region, while other regions may have a greater impact on transportation into the stratosphere.[...]"*

  The final paragraph of the Summary and Conclusion was adapted again, to focus on the probably minor contribution of the investigated events to the stratosphere:

*"By forward FLEXPART calculation of the three events, the potential of these observed elevated CH2Cl2 mixing ratios to reach the tropopause region and lower stratosphere in the following days (12-day time period) was investigated. Even if a substantial proportion reaches the tropopause region (up to 27.6% for flight F17), the contribution to the lower stratosphere is minor for all three events (1.3% to 3.8%). However, it remains unclear whether this observation is limited to the events we observed or if it extends more generally to convection within the East Asian part of the monsoon circulation. More detailed and systematical investigation is needed to determine this."*

- The description about the trajectory experiment configuration is a little unclear to me. Are the trajectories released in a "rectangular prism" shape with dimensions equal to the total latitude, longitude, and altitude spanning by the aircraft during that 5-minute flight segment?  Or are they simply released at the exact location of the flight track?  If the former, what are the dimensions of the "rectangular prism" in number of initialized trajectories?

  In order to make is clearer we have reformulated as follows:

  *"The particles are released in rectangular boxes bounded by the longitude, latitude, and pressure sampled by the aircraft in the respective 5-minute intervals. Within each box, 5000 computational particles are released which are distributed evenly throughout the box. The trajectories of the particles are calculated for 12 days (forward and backward in time). Loss processes due to deposition or chemical reactions are neglected, with transport being the focus only."*

- In the opening paragraphs of Section 3.3, there are several latitude and longitude ranges that are printed in the text for the PBL contacts. I don't think this is particularly insightful without being able to visualize it on a map as given later in Figures 6-8.  I will also add that the first sub-section (3.3.1) has "location" in the title, although some location analysis has already been discussed above it.  I would consider streamlining the location discussion in the text so the map figures can be introduced at the same time.

  Latitude and longitude ranges in section 3.3. are supported by Fig. 3 C and 4 C. This section (3.3) in general is an overview of the events compared to the rest of the other flight sections, whereas sub-section 3.3.1 and 3.3.2 only relate to the events. To make clear that a discussion of the location is not only included in sub section 3.3.1, we have changed the heading of sub section 3.3.1 (see below). However, we would refrain from adding, for

example, the paragraph on locations from section 3.3 to sub-section 3.3.1 to streamline all location discussion to this sub-section.

*"Analysis of the last PBL contacts and the maximum updraft along the trajectories of the events".*

- There are several spots where I believe the word "observed" is misused, in the context of transport or PBL contacts. These processes were not observed by the HALO aircraft, they were simulated by FLEXPART. I suggest going through and changing these instances to be more accurate. I found examples of this on lines 220 and 257, though there may be others.

  We thank the reviewer for pointing this out. We have replaced the misused term in the following places:

  - *Sect 3.3 "The backward trajectories indicate a large variability in the median PBL contacts, and the $CH_2Cl_2$ mixing ratios appear to be sensitive to the last PBL identified by the simulations."*
  - *Sect 3.3.1 "Transport times to the observation location, particularly from northern China, range from approximately 5 to 6 days, with even shorter transport times simulated from trajectories originating in areas near the coast (Fig. 6 B)."*
  - *Sect 3.3.2 "For the second event on flight F08, the greatest updraft, noted in the majority of particles involved [...]"*

**Technical Remarks and Typos:**

- Both "extratropical" and "subarctic" are used in the paper to describe the region that was sampled during PHILEAS. In the abstract (lines 6-7), consecutive sentences use different terms. It might be worth standardizing this term throughout.

  "extratropical" is used in this manuscript together with upper troposphere and lower stratosphere. It is defined broadly as the region poleward of the subtropical jet (e.g. Gettelman 2011). "subarctic" is used here to further narrow down the measurements geographically to the highere latitudes. Generally, subarctic regions fall between 50°N and 70°N latitude.

- Line 9: Change "Asia" to "Asian"

  Done.

- Line 14-15: The parenthetical remark seems out of place given general statements are being made.

  *We have omitted the parentheses and the information.*

- Line 37: I suggest saying the ASMA "confines pollutants". The way the sentence is laid out, "transport barrier" might be harder to visualize for an unfamiliar reader.

  *We have adapted the sentence in line with the review's advice.*

  *"Furthermore, the ASM forms a high pressure system in the UTLS, the Asian Summer Monsoon Anticyclone (ASMA), which confines uplifted pollutants (e.g. Park et al., 2007; Ploeger et al., 2015)."*

- Lines 50 and 55: Would it be better to list the long names for ACCLIP and PHILEAS here rather than waiting for the next section?

  *Full names have been added in parentheses.*

- Section 2.1: The first paragraph is one long sentence. I suggest this section's text just be made into a single paragraph.

  *Done.*

- Line 63: Is it appropriate to define the HALO acronym?

  *We have included the full name of HALO at this position in the manuscript.*

- Line 78: "temporarily" instead of "temporally"?

  *Done.*

- Line 83: Should the instrument acronyms be defined?

  *Done.*

- Line 95: "As for the GhOST" seems out of place

  *We changed "As for" to "Similar to".*

- Line 174: "Major" and "elevated" seem redundant in the section title

  *By eliminating the word "major" in the title, the potential reader might think that we looked into all the elevated events during the campaign.*

Since we focus on the major events of elevated observations, this information should be included in the tile. We changes the tiltle to:

*"Major events of elevated CH2Cl2 in the upper troposphere"*

- Lines 193-196: This technical description about the figure might be better in the caption so it doesn't distract from the analysis.

  We would like to keep this short technical description in the text. In particular, the information on the tailing of the letter plots is subsequently used to explain why we consider F08, but not F16.

- Line 214: What about "The origin of elevated CH2Cl2 events" for this section title?

  We would like to thank the reviewer for this information and change the title accordingly.

- Line 232: I suggest "within the prior 12 days" instead of "in the respective 5 minute intervals"

  Done.

- The first two paragraphs of Section 3.3.2 appear to be mostly technical in nature. I recommend moving all the relevant details back into Section 2 so that Section 3.3.2 can open with science.

  We would like to thank the reviewer for pointing this out. We have streamlined the first paragraph. The technical information has been moved to section 2 and the first sentence has been added to the second paragraph. However, the former second paragraph is important to understand that only snapshots are shown and we would like to keep this information in section 3.2.2.

- Throughout section 3.4, the term "regions" is used to describe different PVU thresholds. I would suggest using "layers" instead to emphasize that these are vertical ranges, whereas I feel "regions" sounds horizontal.

  In section 3.4 and in the discussion and conclusions, the term layer is used instead of region where appropriate.

- Line 337: add "with time" after the word "portion"

  Done.

- Figure 1 caption: the 1-to-1 line is yellow, not red.

> Done.

- Figures 6-8: The text discussions describe transport time in days, but the figure colorbars show transport time in hours. For continuity, I suggest remaking these figures with a unit of days.

   > We apologize for the inconsistencies in the times of the figures and text. Instead of changing the figures, we have added the information in hours to the text and kept the information in days in parantheses.

- Figures 9 and 10: the geopotential height labels are too small to read (if important), and the word "geopotential" appears to be missing from the labeling between the panels. I also suggest "analysis" in the captions instead of "analyse".

   > We have updated these figures with the proposed corrections.

- Figures S5 and S6 are harder to appreciate without the same red "highlights" on the enhanced CH2Cl2 periods like exist in Figures 3 and 4. The discussions between lines 229-237 and lines 252-253 are much harder to interpret without these.

   > We inlcuded the red highlights to these plots.

- It seems that Figures S13-S20 are not referenced in the main body of text. I'm not sure if this is technically an issue or not with the journal, but nonetheless it does make me wonder why they were included with the submission if they don't directly contribute to the analysis.

   > All Figures of the SI are now included in the main text. Fig. S 15 to S 20 are inlcuded for the purpose of completeness, since the figures shown in the manuscript are snapshots from these time intervals. Fig S21 was included in the main text but without excplicit number.